# Transcriptional Regulation of Reproductive Diapause in the Convergent Lady Beetle, *Hippodamia convergens*

**DOI:** 10.3390/insects13040343

**Published:** 2022-03-31

**Authors:** Emily A. W. Nadeau, Melise C. Lecheta, John J. Obrycki, Nicholas M. Teets

**Affiliations:** Department of Entomology, College of Agriculture, Food and Environment, University of Kentucky, Lexington, KY 40546, USA; eana226@uky.edu (E.A.W.N.); melise.lecheta@uky.edu (M.C.L.); john.obrycki@uky.edu (J.J.O.)

**Keywords:** reproductive diapause, Coleoptera, biological control, transcriptomics, RNA-Seq

## Abstract

**Simple Summary:**

Diapause is a dormant period typically controlled by daylength that ensures an insect’s survival through harsh environmental conditions. The convergent lady beetle, *Hippodamia convergens*, undergoes a reproductive diapause in winter, where female ovaries remain immature and no eggs are laid. This species is an important biological control agent, but during diapause, beetles are less likely to eat pest insects. Thus, knowledge of diapause mechanisms may facilitate manipulation thereof to improve biological control. Further, molecular studies of adult diapause and diapause in Coleoptera are relatively lacking. Here, we assembled and annotated a transcriptome for this species and quantified transcript expression changes during diapause. Female beetles were sampled at three times in diapause (early, mid, and late diapause), which allowed us to characterize the molecular processes occurring at distinct transitions throughout diapause. We found that transcripts involved in flight were consistently upregulated during diapause, which is consistent with dispersal flights at this stage, while transcripts involved in ovarian development were downregulated, which is consistent with the shutdown of reproduction in diapausing females. These findings identify key regulators of diapause in *H. convergens* and contribute to a growing body of literature on the molecular mechanisms of diapause across the insect phylogeny.

**Abstract:**

Diapause is an alternate development program that synchronizes an insect’s life cycle with seasonally abundant resources and ensures survival in unfavorable conditions. The physiological basis of diapause has been well characterized, but the molecular mechanisms regulating it are still being elucidated. Here, we present a *de novo* transcriptome and quantify transcript expression during diapause in the convergent lady beetle *Hippodamia convergens*. *H. convergens* is used as an augmentative biocontrol agent, and adult females undergo reproductive diapause that is regulated by photoperiod. We sampled females at three stages (early, mid, and late diapause) and compared transcript expression to non-diapausing individuals. Based on principle component analysis, the transcriptomes of diapausing beetles were distinct from non-diapausing beetles, and the three diapausing points tended to cluster together. However, there were still classes of transcripts that differed in expression across distinct phases of diapause. In general, transcripts involved in muscle function and flight were upregulated during diapause, likely to support dispersal flights that occur during diapause, while transcripts involved in ovarian development were downregulated. This information could be used to improve biological control by manipulating diapause. Additionally, our data contribute to a growing understanding of the genetic regulation of diapause across diverse insects.

## 1. Introduction

Diapause is an alternate development program that synchronizes an insect’s life cycle and improves survival during harsh winter conditions [1,2,3]. Two common characteristics of diapause are arrested development and a suppression of metabolism, and diapause typically happens at a single life stage for a given species [2]. While originally considered a static period with little change, diapause is a dynamic trait that includes distinct phases, with initiation occurring during early diapause (ED), maintenance during mid-diapause (MD), and termination during late diapause (LD) as insects transition to normal development [4]. These distinct eco-physiological phases are supported by recent suborganismal studies, with gene expression studies indicating distinct molecular transitions that occur as diapause progresses [5,6,7,8,9,10]. The ED period generally involves processes for regulating energy storage, upregulating stress response, regulating hormone production, and seeking protection in overwintering sites [11,12,13,14]. During MD, developmental arrest and metabolic suppression are maintained, so processes involved in cell cycle arrest, transcriptional suppression, and energy conservation are often active at this time [15,16,17]. In LD, insects begin progressing towards normal development, and this phase is characterized by an increase in transcription, cellular respiration, metabolism, and hormone signaling [18,19,20]. These physiological hallmarks are typically consistent with changes in gene expression [8], although the molecular regulation of diapause has only been assessed in a select few insect species.

While molecular studies of diapause have increased in recent years [21,22,23,24,25], there is still a lack of consensus on whether conserved elements regulate diapause. For some processes involved in diapause regulation, there appears to be a conserved “toolkit” of genes and pathways across insect orders [26]. For example, insulin and Wnt signaling seem to regulate several aspects of diapause, including developmental and reproductive arrest, extending lifespan, suppressing metabolism, fat hypertrophy and enhancing stress tolerance [27,28]. Canonical circadian rhythm feedback loops are also commonly involved in diapause, likely as a mechanism to sense daylength during diapause induction [29]. Molecular pathways involved in endocrine regulation of diapause, such as ecdysone and/or juvenile hormone signaling, are also commonly observed across diverse diapause programs [1]. Aside from these commonalities, many of the specific molecular processes regulating diapause seem to be unique across species. However, in-depth analysis of the evolutionary physiology of diapause has been hampered by a lack of taxonomic breadth in previous work. The literature has historically been heavily skewed toward Dipterans in pupal diapause; for example, a meta-analysis of diapause transcriptomes included nine Dipteran, one Lepidopteran, and one Hymenopteran species [26]. The accessibility of next generation sequencing has led to a recent increase in diapause transcriptomes in non-Dipteran species, including Hymenoptera, Coleoptera, and Orthoptera [8,30], but several taxa still remain under-sampled.

Here, we assess transcriptomic changes during female overwintering diapause in the convergent lady beetle, *Hippodamia convergens* (Guérin-Méneville) (Coleoptera, Coccinellidae). Diapause in *H. convergens* is associated with arrested ovarian development and lipid accumulation in the fat body [31]. Based on studies of five North American populations, photoperiod was found to be the primary cue for triggering hibernal diapause [32]. Females typically complete a single spring generation while feeding heavily on pest insects, mate with males, and then disperse *en masse* to undergo a winter diapause. Reproductive diapause in winter allows females to conserve energy reserves by avoiding egg-laying. During this overwintering diapause, females need to be mobile enough to find sufficient prey and water resources to ensure their survival until spring [33]. Our study will specifically address the molecular mechanisms of the overwintering diapause that is primarily regulated by photoperiod.

The diapause status of *H. convergens* also influences its efficacy in biological control programs. *H. convergens* is commonly released throughout the US to control pest aphids [34,35,36], and in commercial operations diapausing adults are typically collected *en masse* from overwintering aggregations. When released outside, beetles often show one of the following behaviors: dispersal from the intended field, or limited feeding that prevents effective biological control [36]. Which behavior is observed depends on the diapause state of the aggregations [33], with beetles collected in early-diapause aggregations remaining near the release site but not feeding, while beetles collected in the spring from late-diapause aggregations disperse from the release site prior to feeding. This phenomenon has been a source of frustration for growers, so the ability to manipulate diapause could lead to significant improvements in augmentative biological control. The primary goal of our study is to thoroughly characterize transcriptional changes during diapause in *H. convergens* to identify the regulatory processes that control diapause in this species. Based on previous molecular studies of adult diapause [2,5,8,9], we predicted a downregulation of transcripts involved in ovarian maturation and egg development, an upregulation of transcripts involved in flight and locomotion due to the dispersal flight of the beetles, and differential regulation of insulin signaling transcripts, a common metabolic regulator during diapause. Together, these results may contribute to efforts for improving biological control by providing molecular biomarkers for distinct diapause phases and/or targets for disrupting diapause.

## 2. Materials and Methods

### 2.1. Study Animals

Adult *H. convergens*, collected in Riley County, Kansas 39.19° N, 96.57° W in June 2016, were paired in 0.24 L (8 oz.) paper containers (Choice^®^ brand, Webstaurant Store, Lancaster, PA, USA) covered with organdy mesh. Beetles were maintained at 22 ± 1 °C with a photoperiod of 16:8 L:D and provided water, a Wheast-honey mixture (Wheast is from GreenMethods.com, Beneficial Insectary, Redding, CA, USA), and a daily supply of pea aphids, *Acyrthosiphon pisum* (Harris) (Hemiptera, Aphididae). Egg masses were collected daily from one female and systematically placed in L:D 16:8 or 10:14, at 22 °C ± 1.0 °C. These F1 offspring were individually reared in glass vials at each photoperiod on *A*. *pisum* and frozen *Ephestia kuehniella* (Zeller) (Lepidoptera, Pyralidae) eggs (Beneficial Insectary, Redding, CA, USA).

After eclosion, F1 adults were mated and maintained in 0.24 L paper containers at the same photoperiods, and provided water, Wheast-honey mixture, and a daily supply of pea aphids. The date of first oviposition at L:D 16:8 was recorded, and ovipositing females at L:D 16:8 were flash frozen in liquid nitrogen between 1 and 7 days after the first oviposition and comprised the non-diapause (ND) group. Females at L:D 10:14 that did not oviposit (presumably because they were in reproductive diapause) were frozen at three time periods after female eclosion. Early diapause (ED) females were frozen 7–10 days post-eclosion. This time period is when ND females began laying eggs. Mid-diapause (MD) females were frozen 16–20 days post-eclosion. Late diapause (LD) females were frozen 27–28 days post-eclosion. One female in the late-diapause group (LD-4) was ovipositing at the time of sampling and this sample was removed from subsequent analyses (see below).

### 2.2. RNA Extraction, cDNA Library Preparation, and Sequencing

Each replicate consisted of total body mRNA from a single female, and four independent replicates were sequenced per group (N = 16 RNA-seq libraries in total). RNA was extracted from female beetles using the Direct-Zol RNA Miniprep Kit (Zymo Research, Irvine, CA, USA) according to manufacturer’s instructions. RNA purity and quantity were assessed with UV-VIS spectroscopy, and RNA integrity was measured on a Bioanalyzer (Agilent, Santa Clara, CA, USA). RNA samples were sent to Macrogen (currently Psomagen, Rockville, MD, USA) for mRNA sequencing, and mRNA-seq libraries were sequenced on an Illumina HiSeq 2500 (software version HCS v2.2.38, Illumina, San Diego, CA, USA) to produce 100 bp paired end reads. The number of reads per sample ranged from 33 million to 49 million for a total of 636,203,876 million paired end reads. Each sample was split into left and right read libraries for a total of 32 sequence files. Raw data and the assembled transcriptome from Trinity are available through NCBI, Bioproject # PRJNA753921.

### 2.3. De novo Transcriptome Assembly and Annotation

Before assembly, we analyzed sequence quality with FastQC on Galaxy (Galaxy Version 0.72+galaxy1) [37]. Adapter sequences were removed by the sequencing company prior to transcriptome assembly, and FastQC analysis indicated that there were no issues with per sequence quality scores, sequence duplication scores, or overrepresented sequences. These results indicated that no further quality trimming was necessary. De novo assembly was conducted with the default settings of Trinity v2.4.0 [38,39] on the University of Kentucky Lipscomb (DLX) supercomputer (accession number PRJNA753921). We compared the completeness of the assembled transcriptome to a database of arthropod Benchmark Universal Single Copy Orthologs (BUSCO) using BUSCO v3.0.2 [40]. Transcripts were annotated with Trinotate using BLASTx and BLASTp with an e-value threshold of 1 × 10^−3^ [41]. We also used Trinotate to annotate Gene Ontology (GO) terms [42], Kyoto Encyclopedia of Genes and Genomes (KEGG) terms [43], and protein family (Pfam) domains [44].

### 2.4. Differential Transcipt Expression Analysis

To filter assembly artifacts and poorly supported transcripts, original sequences were mapped back onto the Trinity assembly with Bowtie 2 v2.4.2 [45,46] and reassembled with Cufflinks v2.2.1 [47]. The Cufflinks pipeline reduces the number of transcriptional artifacts, misassembled transcripts, and poorly supported transcripts and increases the statistical support for each remaining transcript [48]. We then quantified transcript abundance for all 16 samples with HTSeq v0.12.4 [49]. Using principal component analysis (PCA) to cluster samples, we identified two outlying samples: MD-2 and LD-4. LD-4 was ovipositing at the time of sampling, which resulted in LD-4 clustering with the nondiapause samples, so we elected to remove it from downstream analyses (see Section 3). A PCA score plot indicated that MD-2 was clearly an outlying sample with likely technical issues, so this sample was also removed from the analyses.

To test for differential transcript expression, we used DESeq2 v1.28.1 on R v.4.0.2 [50,51]. We compared transcript expression between distinct stages of diapause, and the primary comparisons we focused on were ED vs. ND, MD vs. ND, and LD vs. ND. To confirm the results found in these pair-wise comparisons, we also compared All Diapause vs. ND, MD+LD+ND vs. ED, ED+LD+ND vs. MD, and ED+MD+ND vs. LD (Appendix A). Two additional comparisons were made between MD vs. ED and MD vs. LD, to determine how mid- and late diapause differed from the previous diapausing stage. All comparisons were done with alpha = 0.05 after Benjamini-Hochberg correction. To identify biological processes involved in diapause, we performed GO enrichment analysis on differentially expressed transcripts using goseq [52], assuming a Wallenius distribution. A Benjamini-Hochberg correction was applied to raw *p*-values to determine GO terms that were significantly enriched. We used REVIGO with the stringency set to “Medium” to generate a list of nonredundant GO terms [53] and visualize the results. KEGG pathway enrichment was performed with the Generally Applicable Gene-set Enrichment (GAGE) and Pathview packages [54,55]. We accepted pathways as significant if the FDR-adjusted *p*-value was < 0.1.

## 3. Results

### 3.1. Transcriptome Assembly and Annotation

We assembled 636 million paired end, 100-bp reads from 16 libraries into a *de novo* reference transcriptome (see Table 1 for details).

Our assembly contained 96.2% complete BUSCOs when compared to the arthropod database, which is a high degree of completeness compared to other recent *de novo* insect transcriptomes [48,56,57]. The assembly produced 238,614 contigs with a mean contig length of 836 bp, a median contig length of 441 bp, an N50 of 1442 bp, and a GC% of 37.1%. The Trinotate annotation rates were as follows: 31.8% of contigs contained BLAST hits, 29.38% with GO descriptions, 23.9% with KEGG IDs, and 21.93% with Pfam domain(s) (Table 1).

### 3.2. Transcript Expression Changes in Early Diapause

The PCA showed that the ED, MD, and LD samples cluster together with the ND samples forming their own cluster (Figure 1).

To identify transcript expression changes associated with early diapause, we compared the ND and ED groups. In this comparison, there were 1620 upregulated and 2409 downregulated transcripts at FDR < 0.05 (Figure 2a, Appendix A).

GO enrichment analysis indicated a total of 106 enriched GO terms among upregulated transcripts and 116 enriched terms at *p*-value < 0.1 among downregulated transcripts. After clustering semantically redundant terms with REVIGO, most of the enriched BP GO terms in the upregulated transcripts were involved in locomotion (Table 2). For downregulated transcripts, several of the enriched BP GO terms reflect a reduction in protein synthesis and cell division (Table 2). We also ran a contrast to compare gene expression between ED and all the other groups (ND, MD, and LD). Among the genes downregulated in ED relative to all other groups, GO terms related to cell cycle and cell division were similarly enriched (Appendix A).

Between ED and ND, there were 32 significantly upregulated and 19 significantly downregulated KEGG terms at *p*-value < 0.1, and the significantly upregulated and downregulated KEGG term results were qualitatively similar to the enriched GO terms. The top five upregulated KEGG pathways (based on *p*-value) in ED vs. ND were oxidative phosphorylation, cardiac muscle contraction, retrograde endocannabinoid signaling, microbial metabolism in diverse environments, and cGMP-PKG signaling pathway (Table 3). The top five downregulated KEGG pathways in ED vs. ND were RNA transport, cell cycle, DNA replication, ubiquitin mediated proteolysis, and RNA degradation (Table 3). Similar KEGG terms were found when we compared MD+LD+ND vs. ED (Appendix A).

### 3.3. Transcript Expression Changes in Mid-Diapause

Relative to ND samples, there were 912 significantly upregulated and 1758 significantly downregulated transcripts in the MD group relative to ND at FDR < 0.05 (Figure 2b). GO enrichment analysis indicated 83 enriched terms among the upregulated transcripts and 73 enriched terms among the downregulated transcripts. Prominent terms present after REVIGO reduction of GO terms among the upregulated transcripts include muscle contraction, flight, and locomotion (Table 4). Terms of interest among the downregulated transcripts after REVIGO reduction included cell division and cell cycle (Table 4). The list of GO terms was expanded when we compared MD to all other groups simultaneously (ND, ED, and LD) (Appendix A). Protein refolding and regulation of translational initiation by eIF2 alpha phosphorylation were upregulated in MD vs. All Diapause. Negative regulation of stress-activated MAPK cascade, glutaminyl-tRNA aminoacylation, and receptor-mediated endocytosis were downregulated in MD vs. All Diapause.

There were 10 enriched KEGG pathways at *p*-value < 0.1 in the upregulated transcripts. The top five were cardiac muscle contraction, cGMP-PKG signaling pathway, calcium signaling pathway, retrograde endocannabinoid signaling, and phototransduction—fly (Table 5). There were 5 enriched KEGG pathways at *p*-value < 0.1 in the downregulated transcripts: cell cycle, ubiquitin mediated proteolysis, homologous recombination, DNA replication and RNA transport (Table 5). The list of significantly downregulated KEGG terms was expanded when we compared MD to the three other groups simultaneously in a planned contrast (Appendix A, All vs. MD Up OR MD vs. All Down tab). The top five downregulated KEGG terms in this comparison were carbon metabolism, starch and sucrose metabolism, cysteine and methionine metabolism, valine, leucine, and isoleucine degradation, and sphingolipid metabolism.

To further characterize processes involved in mid-diapause, we also compared ED vs. MD (Appendix A). Relative to ED, there were 556 upregulated and 210 downregulated transcripts in the MD group at FDR < 0.05 (Appendix A). GO enrichment analysis indicated 53 enriched terms among the upregulated transcripts and 2 enriched terms among downregulated transcripts (Appendix A, Table 4). A few examples of the upregulated BP terms after REVIGO reduction include polysaccharide catabolic process, lipid transport, and lipid biosynthetic process (Table 4). Only flavin adenine dinucleotide binding and structural constituent of ribosome, which are both MF terms, were significantly enriched in the downregulated transcripts (Appendix A). There were 18 significantly upregulated and 3 significantly downregulated KEGG terms at *p*-value < 0.1. The top 5 upregulated KEGG pathways include biosynthesis of secondary metabolites, starch and sucrose metabolism, oxidative phosphorylation, ABC transporters, and galactose metabolism (Table 5). The three downregulated KEGG pathways are ribosome, RNA transport and cell cycle (Table 5).

### 3.4. Transcript Expression Changes Occurring in Late Diapause

Relative to the ND group, 439 significantly upregulated and 1149 significantly downregulated differentially expressed transcripts at FDR < 0.05 (Figure 2c, Appendix A). There were 36 significantly enriched GO terms among the upregulated transcripts. Terms of interest remaining after REVIGO reduction included locomotion, protein refolding, protein folding chaperone, and misfolded protein binding (Table 6). There were 18 significantly enriched GO terms among the downregulated transcripts (Table 6). Terms of interest remaining after REVIGO reduction included chitin catabolic process, lipid transporter activity, and nutrient reservoir activity. Chitin catabolic process, among others, were also found when we compared LD against all other groups with a planned contrast (Appendix A).

There were no significantly enriched KEGG pathways at *p*-value < 0.1 in the upregulated transcripts. There were ten significantly enriched KEGG pathways in the upregulated transcripts when we compared LD vs. all other groups at *p*-value < 0.1. The top five were ribosome, ribosome biogenesis in eukaryotes, spliceosome, DNA replication, and RNA transport (Appendix A). In the transcripts downregulated in LD relative to ND, there was one significantly enriched KEGG pathway at *p*-value < 0.1, protein processing in endoplasmic reticulum (Table 7). There were 14 significantly enriched KEGG pathways in the downregulated transcripts when we compared LD vs. all other groups (Appendix A). The top five were oxidative phosphorylation, lysosome, phagosome, protein processing in endoplasmic reticulum, and AMPK signaling pathway.

Relative to the MD group, there were 451 upregulated and 411 downregulated differentially expressed transcripts in LD at FDR < 0.05 (Appendix A). There were no significantly enriched GO terms among the up- or downregulated transcripts, and thus no REVIGO reduction was performed. There was one significantly upregulated KEGG term, oxidative phosphorylation, and zero significantly downregulated KEGG terms at *p*-value < 0.1 (Table 5).

## 4. Discussion

Insects enter diapause in advance of unfavorable environmental conditions, often under control of seasonal cues such as changes in photoperiod, humidity, and temperature [4]. Diapause allows insects to synchronize their life cycle with seasonally available resources and survive stressful winter environments [4]. The precise physiological and molecular processes during diapause depend on the life stage in which diapause occurs and the species of interest. Here, we assessed transcript expression changes during diapause in the convergent lady beetle, *H. convergens*, an important biological control agent with an adult reproductive diapause. Our results will be discussed in relation to the phases of diapause, and we will address both conserved processes and novel diapause-associated genes that have not been previously implicated. Identifying putative regulatory transcripts that are expressed at specific points in diapause can generate potential targets for diapause manipulation in future studies.

### 4.1. Changes Occurring in Early and Mid-Diapause

The transcripts upregulated in ED vs. ND and MD vs. ED were enriched for GO terms involved in the cytoskeleton, as well as related terms involved in sarcomere organization, muscle contraction, flight, and locomotion. Upregulation of these transcripts may be connected to two physiological hallmarks of diapause in *H. convergens*: flight dispersal during early diapause [36] and cytoskeletal reorganization to protect against cold-induced disruption of cell structure. *Hippodamia convergens* disperse from feeding fields at the end of the spring to overwinter in the Sierra Nevada Mountains of California [36]. Upregulation of cytoskeletal genes has been documented in another species, *Tetranychus urticae* (Trombidiformes, Tetranychidae), that has similar migratory behavior during its reproductive diapause [24]. Specific transcripts upregulated relative to ND in *H. convergens* included components of muscle filaments such as titin (GJNK01217174, log_2_FC = 1.18, *p* = 2.84 × 10^−4^, where the accession number is for the NCBI nucleotide database, “log_2_FC” is the log2 fold change, and the *p*-value is the FDR-adjusted *p*-value), myosin (both heavy [GJNK01010876, log_2_FC = 1.18, *p* = 5.59 × 10^−3^] and light chains [GJNK01120126, log_2_FC = 1.67, *p* = 1.46 × 10^−3^]), troponin (GJNK01183047, log_2_FC = 1.32, *p* = 1.08 × 10^−2^) and tropomyosin (GJNK01150561, log_2_FC = 1.27, *p* = 4.81 × 10^−3^). Additional muscle-related transcripts upregulated during diapause include muscle LIM protein (GJNK01162820, log_2_FC = 1.42, *p* = 4.78 × 10^−3^), which includes a zinc-binding motif that regulates cytoarchitecture and signal transduction for gene expression, and muscle-specific protein 20 *mlp20* (GJNK01192324, log_2_FC = 1.23, *p* = 1.19 × 10^−2^), a calponin-like actin binding domain that cross-links actin filaments and titin molecules in the Z-disc [58,59,60]. In addition, nearly all of these genes were also upregulated in ED relative to all other groups in a planned contrast (Appendix A), indicating that they are specific to ED. In addition to supporting increased flight activity during diapause, cytoskeletal genes are also linked to increased cold tolerance during diapause by increasing the structural stability of cells to enhance survival at low temperatures [24,48,61,62,63,64,65,66].

While metabolic suppression is often a feature of diapause, we observed upregulation of oxidative phosphorylation transcripts during ED and MD (seen after KEGG enrichment analysis, Table 3 and Table 5). Diapause is typically accompanied by a decrease in metabolism and therefore a downregulation of oxidative phosphorylation to conserve energy for post-diapause processes such as metamorphosis or migratory flight [17]. However, *H. convergens* actively flies during its reproductive diapause and therefore has high energy demands. Thus, increased expression of oxidative phosphorylation transcripts may be required to meet the metabolic demands of the dispersal flight. However, it is worth noting that a similar pattern of oxidative phosphorylation gene expression was observed in the Colorado potato beetle which spends diapause underground [67]. Therefore, these changes in transcript expression may be unrelated to flight and instead a general feature of adult reproductive diapause in Coleoptera.

GO and KEGG terms related to the cell cycle were significantly downregulated in both the ED vs. ND and MD vs. ED comparisons. This downregulation of cell cycle transcripts is consistent with previous observations that cell division is reduced or stopped during diapause [15,16,68,69,70]. A recent study by Shimizu et al. (2018) showed clear evidence that pupal diapause in *Nasonia vitripennis* (Hymenoptera, Pteromalidae) caused the disappearance of the S fraction of the cell cycle and a high proportion (80%) of cells arrested in the G0/G1 phase. In *H. convergens*, the cell cycle arrest likely reflects the halt in ovarian development during reproductive diapause to prevent egg maturation. We also observed many core ribosomal protein transcripts downregulated in ED relative to ND, including *60S ribosomal protein L32* (GJNK01022121, log_2_FC = −23.21, *p* = 2.58 × 10^−12^) and *60S ribosomal protein L21* (GJNK01130154, log_2_FC = −22.24, *p* = 2.57 × 10^−11^), and both of these were also downregulated in ED relative to all other stages in a planned contrast. When comparing ED to all other stages, *40S ribosomal protein S2* (GJNK01061680, log_2_FC = −0.43, *p* = 6.12 × 10^−3^), was also downregulated, and this gene has previously been linked to slowed and halted cell growth [71]. Diapause in the lady beetle *Coccinella septempunctata* (Coleoptera, Coccinellidae) also appears to be accompanied by cell cycle arrest, as several genes involved in cell cycle are downregulated during diapause termination [22]. In particular, DNA replication licensing factor *MCM6* was downregulated in *C. septempunctata*, and while this specific gene was not downregulated in *H. convergens*, the licensing factor *MCM3* (GJNK01201780, log_2_FC = −0.70, *p* = 3.57 × 10^−2^), which is a part of the MCM2-7 complex along with *MCM6*, was downregulated in MD relative to ND. This complex is essential for DNA replication and the cell cycle [72], so downregulation of members of this complex likely prevents cell division. *Cyclin A/B*, which activates cyclin-dependent kinases at various points in the cell cycle to progress cell division [73,74] is also downregulated during diapause in *C. septempunctata* [22]. While this specific gene wasn’t downregulated in our study, *Cyclin-A1*, another A-type cyclin that contributes to the G1 to S cell progression [75], was downregulated at all three diapause time points relative to ND.

During normal development in many insect species, juvenile hormone (JH) induces vitellogenesis and ovarian maturation [76,77]. Indeed, several transcripts encoding *vitellogenin-1*, one of the main yolk proteins, were strongly downregulated during ED and MD, which is consistent of a shutdown in ovarian maturation. Transcript expression data also indicated an involvement of juvenile hormone in this ovarian arrest. Juvenile hormone esterases can control the JH titer by hydrolyzing JH and leading to its degradation [78]. In *H. convergens*, *juvenile hormone esterase* (JHE) had lower expression in LD compared to MD (GJNK01138836, log_2_FC = −24.70, *p* = 4.57 × 10^−16^), which is consistent with JH signaling being activated during LD as the beetles resume reproduction. Thus, low expression of JHE during late diapause likely facilitates termination of ovarian arrest, and this pattern of JHE was also observed during the reproductive diapause of *Galeruca daurica* (Coleoptera, Chrysomelidae) [30].

In many insects, JH mediated vitellogenesis is regulated upstream by insulin signaling [79,80]. *Culex pipiens* (Diptera, Culicidae), which has perhaps the best characterized adult diapause, shuts down insulin signaling during diapause which in turn suppresses JH synthesis and leads to the activation of FOXO, a transcription factor involved in fat hypertrophy and stress tolerance [77]. The suppression of JH synthesis in turn prevents ovarian maturation. Because diapause in *H. convergens* is accompanied by arrested ovarian development, we predicted that diapause would result in downregulation of transcripts involved in both insulin and JH signaling [27]. Indeed, *insulin-like growth factor-binding protein (IGFBP) complex acid labile subunit* is downregulated during MD relative to both ND (GJNK01122314, log_2_FC = −7.61, *p* = 2.42 × 10^−2^) and all other groups (log_2_FC = −8.06, *p* = 5.2 × 10^−3^). IGFBPs bind to members of the insulin superfamily, which include insulin-related peptides (IRPs) and insulin-like peptides (ILPs), and they positively regulate ecdysteroidogenesis, JH synthesis, and vitellogenesis [79,80]. Thus, downregulation of this transcript in MD is consistent with suppression of ovarian development and vitellogenesis in *H. convergens*. We did observe downregulation of another insulin-related transcript, *insulin-like peptide receptor*, during ED and LD relative to ND (GJNK01203393, log_2_FC = −0.83, *p* = 5.10 × 10^−3^ for ED; log_2_FC = −0.73, *p* = 1.57 × 10^−2^ for LD). In mosquitoes, suppression of insulin signaling leads to an upregulation of FOXO to promote fat hypertrophy. In the mosquito *C. pipiens*, *forkhead box protein O3 (FOXO3*) is upregulated in diapause [77], and consistent with this model *FOXO3* is upregulated during diapause in the lady beetle *C. semptempunctata*, which likely results in fat hypertrophy observed in this species [22]. However, we did not see any evidence of FOXO being involved in diapause in *H. convergens*.

While the above results indicate a possible role for insulin and JH in *H. convergens* diapause, many transcripts involved in insulin and JH signaling did not change expression during diapause. One possible reason is that we only sampled adults. Although we do not have direct evidence for *H. convergens*, a study on the closely related *Hippodamia variegata* (Coleoptera, Coccinellidae) determined that the photosensitive stage is the pupal stage [81], so if insulin and/or JH are involved in diapause induction, expression may have changed prior to our sampling interval. Further, we also sampled whole bodies, instead of tissues directly related to insulin-signaling such as the fat body and the brain, which could have prevented us from observing expression changes in these transcripts [67,82]. Likely, for these same reasons, we didn’t detect any evidence of circadian clock genes being involved in *H. convergens* diapause, despite their prominent role in other species [29].

Our results point to the cGMP-Protein Kinase G (PKG) pathway as a potential novel regulator of diapause (Table 3 and Table 5). This KEGG pathway for cGMP-PKG was upregulated in both ED and MD relative to ND, and this pathway has established roles in increasing survival during anoxic stress [83], as well as being a key pathway for associative learning and memory [84] and food search behavior [85]. A proteomics analysis linked cGMP-PKG signaling to pupal diapause termination in the fly *Bactrocera minax* (Diptera, Tephritidae), although there was no evidence of transcriptional regulation of this pathway [86]. Because signaling pathways like cGMP-PKG are often primarily regulated at the post-translational level through reversible protein modifications, it’s difficult to conclude a definitive role for this pathway in reproductive diapause in *H. convergens*. However, coordinated changes in transcript expression across this pathway suggest this pathway are consistent with cGMP-PKG being involved in diapause regulation.

### 4.2. Changes Occurring in Late Diapause

In contrast to the other comparisons, many of the transcripts involved in LD were not annotated and thus have unclear function. This result could suggest that processes occurring in LD are more species-specific and less conserved than other diapause time points. As discussed above, we did see evidence of juvenile hormone involvement in LD, as *juvenile hormone esterase* (GJNK01138836) was downregulated in late diapause, presumably to facilitate accumulation of JH as oogenesis resumes. However, for the most part, transcripts involved in LD were sparsely annotated, which was also reflected by lack of enriched GO and KEGG terms at this time point.

One somewhat surprising result is that we observed strong upregulation of *hsp68* in LD relative to ND (GJNK01155541, log_2_FC = 1.22, *p* = 2.48 × 10^−9^), and this transcript was also upregulated in LD relative to all other time points. Heat shock proteins (HSPs) are often upregulated in direct response to abiotic stress, but during diapause there is often anticipatory upregulation of HSPs that occurs before any environmental stress occurs [87]. However, while a role for HSPs in diapause is well-established, the specific HSP genes involved in diapause varies considerably across species [88,89,90,91]. While we expected HSP expression to return to baseline in LD, at this time *H. convergens* leave their aggregations and may still experience periodic low temperatures. Thus, we speculate that upregulation in *hsp68* may be needed to withstand environmental stress as beetles disperse from aggregations, and indeed, this gene is known to be important for surviving low temperatures [92]. It is worth noting that different HSP transcripts were upregulated in ED (e.g., *hsc70* [GJNK01113255, log_2_FC = 8.32, *p* = 1.10 × 10^−2^], *hsp70 A2* [GJNK01189679, log_2_FC = 2.12, *p* = 3.88 × 10^−2^] and *small heat shock protein C1* [GJNK01016550, log_2_FC = 1.17, *p* = 9.69 × 10^−4^]), but taken together, our results are consistent with previous observations that HSP expression patterns during diapause tend to be species- and stage-specific.

### 4.3. Practical Applications and Future Directions

Diapause complicates the use of *H. convergens* as a biocontrol agent because females have different feeding behaviors based on their diapause phase. Further, the diapause trajectory can vary across populations, across individual aggregations, and even across beetles within a single aggregation [32]. The information provided in this study will permit in-depth characterizations of diapause variation in future work and may also be used to manipulate diapause for applied purposes.

We envision two possible benefits of diapause manipulation in *H. convergens*: extending the shelf life of stored beetles and breaking diapause so that beetles continue to eat pest insects. In *C. septempunctata*, diapausing females have a lifespan of more than 220 days, compared to approximately 60 days in non-diapausing beetles. Thus, the ability to artificially promote diapause and/or prevent diapause termination could improve the long-term storage capacity of beetles. Conversely, the ability to break diapause could lead to increased predation at times of year when beetles are normally dormant, and our results could be used to develop transcriptional biomarkers for distinct phases of diapause to determine the developmental stage of beetles collected in the field. For example, there are several genes with distinct expression patterns in MD that could be used to assess whether a population of beetles is in the maintenance phase of diapause. Some of these genes include *nucleosome assembly protein 1-like 4*, which is downregulated in MD relative to ND and all other groups (GJNK01191961, log_2_FC = −1.86, *p* = 8.68 × 10^−8^ vs. ND; log_2_FC = −1.27, *p* = 4.35 × 10^−5^ vs. all other groups) and *divergent protein kinase domain 1C*, which is upregulated in MD relative to ND and all other groups (GJNK01027417, log_2_FC = 2.80, *p* = 5.04 × 10^−4^ vs. ND; log_2_FC = 1.94, *p* = 7.42 × 10^−3^ vs. all other groups). This idea has been previously proposed for *C. septempunctata* [22], which has similar reproductive diapause as *H. convergens*, although to our knowledge, these ideas have not been put into practice. Diapause manipulation using molecular means has been successfully achieved for some pest insects, including the red palm weevil [93] and corn earworm [94], and similar strategies could be applied to beneficial insects. However, these strategies require a thorough understanding of the molecular regulation of diapause, and our study provides essential information on the processes regulating distinct phases of diapause in *H. convergens*.

## Figures and Tables

**Figure 1 insects-13-00343-f001:**
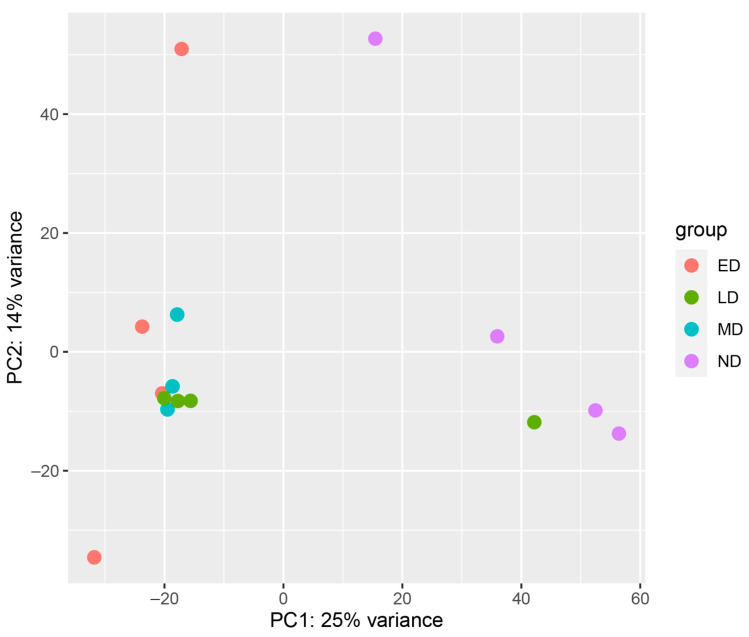
Principle Component Analysis (PCA) score plots of samples. This PCA was used to determine the relationship between transcript expression profiles from various stages of diapause. One of the mid-diapause samples was removed from the dataset prior to this analysis due to technical problems with the sample, so there are only three replicates for this group. One LD sample clustered with the ND samples, and that female had already broken diapause and started ovipositing at the time of sample collection. Thus, this sample was removed from further analyses. ED = early diapause, LD = late diapause, MD = mid-diapause, ND = non-diapause.

**Figure 2 insects-13-00343-f002:**
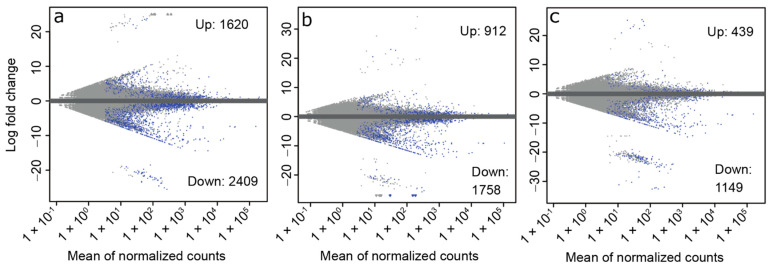
MA plots showing the number of differentially expressed transcripts between each diapausing stage and non-diapause beetles. For each comparison ((**a**): ED vs. ND; (**b**): MD vs. ND; (**c**): LD vs. ND), transcripts with FDR < 0.05 are shown in blue. In each panel the total number of up- and downregulated transcripts are indicated. For the full lists of differentially expressed transcripts, see Appendix A. ED = early diapause, MD = mid-diapause, LD = late diapause, ND = non-diapause.

**Table 1 insects-13-00343-t001:** *De novo* assembly and annotation statistics. 636 million paired-end reads were assembled into 238,614 contigs with Trinity. BUSCOs were compared with all other available Arthropod transcriptomes. BLAST, GO, KEGG, and Pfam domains were annotated using Trinotate.

Assembly Statistics		Annotation Statistics	
Assembly length (bp)	199,457,684	Complete BUSCOs	96.2%
Contigs	238,614	Contigs with BLAST hit	31.8%
N50	1442	Contigs with GO description	29.4%
Range of contig lengths	201–17,262	Contigs with KEGG IDs	23.9%
Number of reads	636,203,876	Contigs with Pfam domain(s)	21.9%

**Table 2 insects-13-00343-t002:** Summary of the GO biological processes that are involved in ED. The table shows GO terms that are significantly enriched among transcripts that are either up- or downregulated during ED relative to ND. The top five up and down enriched GO terms for this comparison are shown. Redundant GO terms were removed using semantic clustering with REVIGO. GO terms that are enriched in the MD vs. ED comparison are shown in Table 3, and a complete list of enriched GO terms can be found in Appendix A.

Comparison	GO ID	GO Term	Unadjusted *p*-Value	Adjusted *p*-Value
Upregulated Relative to ND	GO:0045214	Sarcomere organization	1.49 × 10^−14^	1.08 × 10^−10^
	GO:0030239	Myofibril assembly	2.54 × 10^−9^	6.16 × 10^−6^
	GO:0007517	Muscle organ development	1.91 × 10^−8^	3.97 × 10^−5^
	GO:0030241	Skeletal muscle myosin thick filament assembly	2.19 × 10^−8^	3.98 × 10^−5^
	GO:0006936	Muscle contraction	2.95 × 10^−8^	4.77 × 10^−5^
Downregulated Relative to ND	GO:0007049	Cell cycle	3.54 × 10^−13^	2.58 × 10^−9^
	GO:0051301	Cell division	1.73 × 10^−12^	4.20 × 10^−9^
	GO:0006417	Regulation of translation	4.59 × 10^−8^	4.45 × 10^−5^
	GO:0005980	Glycogen catabolic process	1.39 × 10^−7^	1.12 × 10^−4^
	GO:0005978	Glycogen biosynthetic process	2.25 × 10^−7^	1.72 × 10^−4^

**Table 3 insects-13-00343-t003:** Summary of the top five (when available) KEGG pathways involved in ED. The table shows KEGG pathways that were up- and downregulated relative to ED. Enriched KEGG pathways relative to ND were identified with the GAGE and Pathview packages in R. Terms were sorted by the Benjamini-Hochberg adjusted *p*-value. KEGG pathways differentially expressed relative to MD are shown in Table 5, and a complete list of up- and downregulated KEGG terms can be found in Appendix A.

Comparison	KO ID	KEGG Term	Adjusted *p*-Value
Upregulated Relative to ND	ko00190	Oxidative phosphorylation	1.23 × 10^−7^
	ko04260	Cardiac muscle contraction	1.66 × 10^−7^
	ko04723	Retrograde endocannabinoid signaling	7.58 × 10^−5^
	ko01120	Microbial metabolism in diverse environments	9.00 × 10^−5^
	ko04022	cGMP-PKG signaling pathway	2.18 × 10^−3^
Downregulated Relative to ND	ko03013	RNA transport	2.13 × 10^−7^
	ko04110	Cell cycle	2.13 × 10^−7^
	ko03030	DNA replication	1.02 × 10^−6^
	ko04120	Ubiquitin mediated proteolysis	2.95 × 10^−5^
	ko03018	RNA degradation	6.92 × 10^−4^

**Table 4 insects-13-00343-t004:** Summary of the GO biological processes that are involved in MD. The table shows GO terms that are significantly enriched in differentially expressed transcripts from the MD vs. ND, MD vs. ED, and MD vs. LD comparisons. The top five enriched GO terms among up- and downregulated transcripts for these comparisons are shown. Redundant GO terms were removed using semantic clustering with REVIGO. A complete list of enriched GO terms can be found in Appendix A.

Comparison	GO ID	GO Term	Unadjusted *p*-Value	Adjusted *p*-Value
Upregulated Relative to ND	GO:0045214	Sarcomere organization	5.10 × 10^−14^	2.47 × 10^−10^
	GO:0030241	Skeletal muscle myosin thick filament assembly	4.21 × 10^−10^	1.23 × 10^−6^
	GO:0007517	Muscle organ development	1.87 × 10^−9^	4.54 × 10^−6^
	GO:0060537	Muscle tissue development	8.28 × 10^−9^	1.42 × 10^−5^
	GO:0015866	ADP transport	8.75 × 10^−9^	1.42 × 10^−5^
Downregulated Relative to ND	GO:0043161	Proteasome-mediated ubiquitin-dependent protein catabolic process	3.60 × 10^−7^	1.05 × 10^−3^
	GO:0051301	Cell division	3.18 × 10^−6^	6.61 × 10^−3^
	GO:0010498	Proteasomal protein catabolic process	4.20 × 10^−6^	6.77 × 10^−3^
	GO:0008103	Oocyte microtubule cytoskelton polarization	6.66 × 10^−6^	6.77 × 10^−3^
	GO:0007052	Mitotic spindle organization	6.95 × 10^−6^	6.77 × 10^−3^
Upregulated Relative to ED	GO:0000272	Polysaccharide catabolic process	1.51 × 10^−7^	7.33 × 10^−4^
	GO:0006032	Chitin catabolic process	1.65 × 10^−6^	2.67 × 10^−3^
	GO:0006869	Lipid transport	5.46 × 10^−6^	7.23 × 10^−3^
	GO:0015986	ATP synthesis coupled proton transport	9.58 × 10^−6^	1.09 × 10^−2^
	GO:0007594	Puparial adhesion	9.78 × 10^−6^	1.09 × 10^−2^
Downregulated Relative to ED		None		
Upregulated Relative to LD		None		
Downregulated Relative to LD		None		

**Table 5 insects-13-00343-t005:** Summary of the top five (when available) KEGG pathways involved in MD. The table shows KEGG pathways that were up- or downregulated relative to ND, ED, and LD. Enriched KEGG pathways were identified with the GAGE and Pathview packages in R. Terms were sorted by the Benjamini-Hochberg adjusted *p*-value. A complete list of up- and downregulated KEGG terms can be found in Appendix A.

Comparison	KO ID	KEGG Term	Adjusted *p*-value
Upregulated Relative to ND	ko04260	Cardiac muscle contraction	4.30 × 10^−3^
	ko04022	cGMP-PKG signaling pathway	4.30 × 10^−3^
	ko04020	Calcium signaling pathway	1.87 × 10^−2^
	ko04723	Retrograde endocannabinoid signaling	3.40 × 10^−2^
	ko04745	Phototransduction—fly	5.20 × 10^−2^
Downregulated Relative to ND	ko04110	Cell cycle	4.63 × 10^−2^
	ko04120	Ubiquitin mediated proteolysis	4.63 × 10^−2^
	ko03440	Homologous recombination	5.15 × 10^−2^
	ko03030	DNA replication	5.15 × 10^−2^
	ko03013	RNA transport	5.15 × 10^−2^
Upregulated Relative to ED	ko01110	Biosynthesis of secondary metabolites	6.47 × 10^−6^
	ko00500	Starch and sucrose metabolism	2.95 × 10^−4^
	ko00190	Oxidative phosphorylation	6.35 × 10^−4^
	ko02010	ABC transporters	2.68 × 10^−3^
	ko00052	Galactose metabolism	2.85 × 10^−3^
Downregulated Relative to ED	ko03010	Ribosome	1.04 × 10^−3^
	ko03013	RNA transport	8.02 × 10^−2^
	ko04110	Cell cycle	8.02 × 10^−2^
Upregulated Relative to LD	ko00190	Oxidative phosphorylation	8.98 × 10^−3^
Downregulated Relative to LD		None	

**Table 6 insects-13-00343-t006:** Summary of the GO biological processes that are involved in LD. The table shows GO terms that are significantly enriched among transcripts that are either up- or downregulated during LD relative to ND. The top five up and down enriched GO terms for this comparison are shown. Redundant GO terms were removed using semantic clustering with REVIGO. GO terms that are enriched in the MD vs. LD comparison are shown in Table 4, and a complete list of enriched GO terms can be found in Appendix A.

Comparison	GO ID	GO Term	Unadjusted *p*-Value	Adjusted *p*-Value
Upregulated Relative to ND	GO:0045214	Sarcomere organization	1.24 × 10^−9^	9.02 × 10^−6^
	GO:0060537	Muscle tissue development	3.88 × 10^−9^	1.88 × 10^−5^
	GO:0007517	Muscle organ development	5.52 × 10^−9^	2.01 × 10^−5^
	GO:0042742	Defense response to bacterium	4.72 × 10^−7^	8.59 × 10^−4^
	GO:0034620	Cellular response to unfolder protein	3.65 × 10^−6^	4.23 × 10^−3^
Downregulated Relative to ND	GO:0048065	Male courtship behavior, veined wing extension	1.07 × 10^−6^	2.60 × 10^−3^
	GO:0006032	Chitin catabolic process	1.33 × 10^−5^	2.15 × 10^−2^
	GO:0048082	Regulation of adult chitin-containing cuticle pigmentation	1.52 × 10^−5^	2.21 × 10^−2^
	GO:0048067	Cuticle pigmentation	2.43 × 10^−5^	3.08 × 10^−2^
	GO:0000272	Polysaccharide catabolic process	2.54 × 10^−5^	3.08 × 10^−2^

**Table 7 insects-13-00343-t007:** Summary of the top five (when available) KEGG pathways involved in LD. The table shows pathways up- and downregulated relative to ND. Enriched KEGG pathways were identified with the GAGE and Pathview packages in R. Terms were sorted by the Benjamini-Hochberg adjusted *p*-value. Pathways differentially expressed relative to MD are shown in Table 5, and a complete list of up- and downregulated KEGG terms can be found in Appendix A.

Comparison	KO ID	KEGG Term	Adjusted *p*-Value
Upregulated Relative to ND		None	
Downregulated Relative to ND	ko04141	Protein processing in endoplasmic reticulum	4.14 × 10^−2^

## Data Availability

Data is available at https://www.ncbi.nlm.nih.gov/bioproject/753921 under accession number PRJNA753921. Data were published on 11 August 2021.

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
