# Peer review of "Transcriptional Regulation of Reproductive Diapause in the Convergent Lady Beetle, Hippodamia convergens"

_insects, 2022, doi:10.3390/insects13040343_

Round 1

Reviewer 1 Report

The authors present a transcriptomic study on three stages of reproductive diapause in the convergent lady beetle, which is an important biological control of crop pests. The manuscript is well written, the results are generally well presented. The authors present some interesting findings on important transcripts during each of the three phases of diapause. These results are really well presented and discussed. However, I have one major issue with the transcriptome assembly and a few suggestions for improving the results section.

Major concern:

Before I move on to a couple of suggestion, which I think can help improve the results section, there is one rather major concern regarding the transcriptome assembly:
The methods are very well presented, with high detail and well written. Most of the methods seem sound and rigorous, except that it appears that the authors did not perform read trimming prior to transcriptome assembly. Quality and adapter trimming is not necessary for gene expression analyses but is important for de novo assembly to remove low quality bases and remnants of adapters. If this was not performed, it could lower the quality of the transcriptome assembly or cause artifacts.
Therefore, unfortunately, I believe that the filtering/trimming step should be performed and the assembly repeated.

Minor comments:
1. When working with a de novo transcriptome with such a high number of contigs (>200k), these sequences should be referred to as transcripts rather than genes. I suggest to only use the term transcript throughout the manuscript, except in the discussion where results from other studies are being reported.

2. The classification into "up-" and "down-"regulated transcripts (e.g. tables 2-5) is not intuitive and can be confusing. "Up" in MD vs LD, for example, is also "down" in LD, and vice versa - it is a matter of perspective. It would be better to formulate these comparisons more explicitly, as on lines 261-262. It may make more sense to split the tables into ED (up and down vs ND), MD (vs ED and ND) and LD (vs MD and ND). This could help with the presentation and interpretation of the results.

3. It would also be interesting to see significant transcripts, enriched GO and KEGG terms when comparing each of the three stages against all others, i.e. ED vs "ND&MD&LD", to find terms more specific to each of the stages. The current method of comparing each to ND is likely to contain a high overlap of trancripts that are important for the whole diapause. These could also be included in the same tables mentioned in the last point.

4. It might be helpful to present some significant transcripts in the results section, rather than just presenting them in a supplementary table. For example, the top 10 most significant transcripts when comparing ED,MD and LD against ND or against all others.

Reviewer 2 Report

Transcriptional regulation of reproductive diapause in the convergent lady beetle, Hippodamia convergens

The authors take up an interesting topic about the reproductive diapause in the convergent lady beetle Hippodamia convergens. The work is carefully prepared and well written. In particular, the Discussion is well thought out. I have only a few comments to improve this work:

  1. The information given in the Introduction indicates that diapause in all insects is a complete developmental arrest, while this process can take the form of "diapause development", which means slow progression of development during diapause, so please mention this.
  2. Please use the full species name for the first time and then convergens consistently.
  3. In my opinion, both Table 1 and Figure 1 should be part of the authors' results, not the methodology.
  4. Please rewrite sentence 442-444 as it suggests diapause is a response to direct environmental stress and not a programmed phenomenon.
  5. The section Conclusion is rather part of the Discussion. In this chapter, I ask the authors to propose and provide information on the regulation of which genes can be good markers of mid-diapause (MD, diapause maintenance, the most important part of diapause), helpful in diagnosing the activity of various insects. This can be very helpful in determining if an insect is going into diapause.

Round 2

Reviewer 1 Report

All of my concerns have been addressed. I think this is a sound, well-written study.